# ARCHITECT: Generating Vivid and Interactive 3D Scenes with Hierarchical 2D Inpainting

**Yian Wang**[*]
Umass Amherst
yianwang@umass.edu

**Xiaowen Qiu**[*]
Umass Amherst
xiaowenqiu@umass.edu

**Jiageng Liu**[*]
Umass Amherst
jiagengliu@umass.edu

**Zhehuan Chen**
Umass Amherst
zhehuanchen@umass.edu

**Jiting Cai**
Shanghai Jiao Tong University
caijiting@sjtu.edu.cn

**Yufei Wang**
Carnegie Mellon University
yufeiw2@andrew.cmu.edu

**Tsun-Hsuan Wang**
MIT
tsunw@mit.edu

**Zhou Xian**
Carnegie Mellon University
zhouxian@cmu.edu

**Chuang Gan**
Umass Amherst
chuanggan@umass.edu

## Abstract

Creating large-scale interactive 3D environments is essential for the development of Robotics and Embodied AI research. However, generating diverse embodied environments with realistic detail and considerable complexity remains a significant challenge. Current methods, including manual design, procedural generation, diffusion-based scene generation, and large language model (LLM) guided scene design, are hindered by limitations such as excessive human effort, reliance on predefined rules or training datasets, and limited 3D spatial reasoning ability. Since pre-trained 2D image generative models better capture scene and object configuration than LLMs, we address these challenges by introducing ARCHITECT, a generative framework that creates complex and realistic 3D embodied environments leveraging diffusion-based 2D image inpainting. In detail, we utilize foundation visual perception models to obtain each generated object from the image and leverage pre-trained depth estimation models to lift the generated 2D image to 3D space. While there are still challenges that the camera parameters and scale of depth are still absent in the generated image, we address those problems by "controlling" the diffusion model by *hierarchical inpainting*. Specifically, having access to ground-truth depth and camera parameters in simulation, we first render a photo-realistic image of only back-grounds in it. Then, we inpaint the foreground in this image, passing the geometric cues in the back-ground to the inpainting model, which informs the camera parameters. This process effectively controls the camera parameters and depth scale for the generated image, facilitating the back-projection from 2D image to 3D point clouds. Our pipeline is further extended to a hierarchical and iterative inpainting process to continuously generate placement of large furniture and small objects to enrich the scene. This iterative structure brings the flexibility for our method to generate or refine scenes from various starting points, such as text, floor plans, or pre-arranged environments. Experimental results demonstrate that ARCHITECT outperforms existing methods in producing realistic and complex environments, making it highly suitable for Embodied AI and robotics applications.[2]

---

[*]Equal Contribution

[2]Project page: https://vis-www.cs.umass.edu/ARCHITECT

38th Conference on Neural Information Processing Systems (NeurIPS 2024).

# 1 Introduction

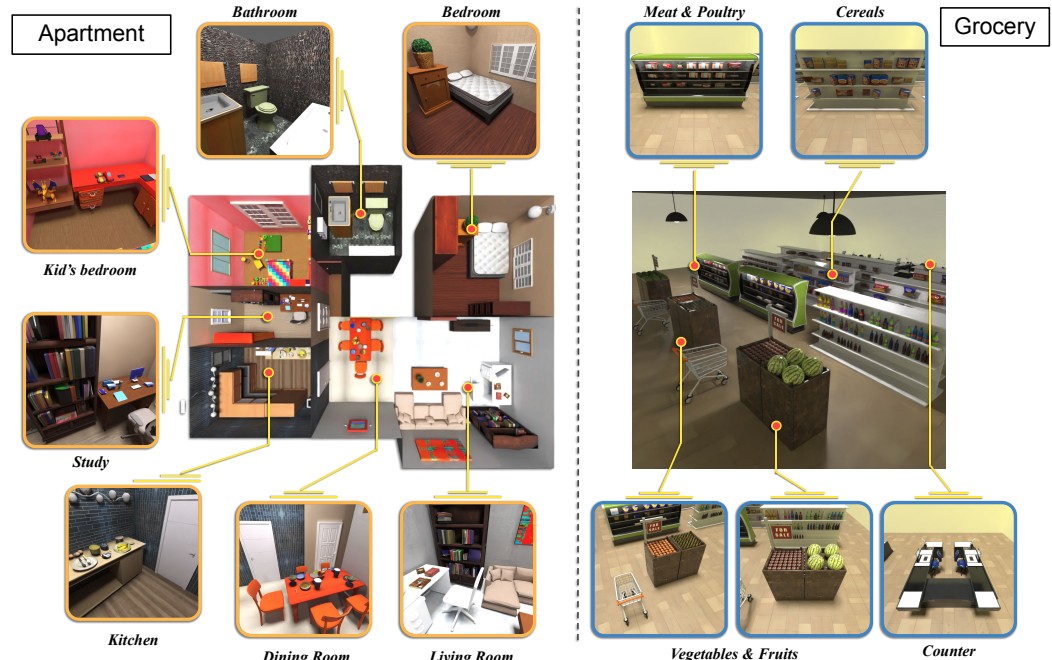

Figure 1: We present ARCHITECT, a generative framework to create *diverse*, *realistic*, and *complex* Embodied AI scenes. Leveraging 2D diffusion models, ARCHITECT generates scenarios in an open-vocabulary manner. Here, we showcase two cases in detail: an apartment and a grocery store.

Collecting or generating large-scale training data has recently emerged as a promising direction for advancing Robotics and Embodied AI research. A major focus in recent works pursuing this direction advocates for data generation in simulated environments [Wang et al., 2023b,a, Ha et al., 2023, Dalal et al., 2023], as simulation offers a cost-effective approach to data collection that scales naturally with computational resources; this thrust holds the potential for producing realistic physics and rendering data, and meanwhile grants access to valuable ground-truth state information for speeding up policy learning. Among the types of data needed for training Embodied AI agents, diverse and realistic *environments* with the possibility of interacting with surrounding entities is crucial. However, obtaining vivid interactive scene and environment data remains challenging. Recent studies have attempted to tackle this problem by developing generative models for environment creation via various approaches, including procedural generation with predefined rules [Deitke et al., 2022], diffusion-based scene generation [Tang et al., 2023a, Yang et al., 2024b, Feng et al., 2024], and large language model (LLM) guided scene population and design [Wang et al., 2023b, Yang et al., 2024c, Wen et al., 2023].

Despite these recent efforts, generating *diverse*, *realistic*, and *complex* Embodied AI environments still remains a challenging problem due to the inherent drawbacks and assumptions made in the pipeline designs of existing methods. For example, manually designed environment datasets [Ramakrishnan et al., 2021, Weihs et al., 2021, Li et al., 2023a, Fu et al., 2020a,b] require excessive human effort and are hence inherently difficult to scale. Procedural generation methods [Deitke et al., 2022, Khalifa et al., 2020, Earle et al., 2021, Zhao et al., 2021] rely on predefined rules, which are limited in their ability to learn from and resemble real-world distributions, and struggle to generate open-vocabulary scenes. Large language model (LLM) guided scene generation process [Wang et al., 2023b, Yang et al., 2024c, Wen et al., 2023, Lin et al., 2023, Aguina-Kang et al., 2024, Feng et al., 2024] also presents its own limitations, as LLMs operate in language space and have limited 3D understanding and spatial reasoning capabilities. Moreover, existing LLM-based scene generation methods still rely on certain simplifications and hand-designed rules, such as primarily focusing on placements of large furniture pieces on the floor or against walls, and only considering simple inter-object relationships such as random placement of small items *on top of* large background furniture. Therefore, these methods struggle to generate more complex and cluttered object arrangements that are often encountered in

daily life, such as "an organized dining table", "an office desk drawer cluttered with objects", or "a shelf of toys", which typically require nuanced object placement and context-aware positioning.

As a result, there still exists a gap in current literature for generating interactive 3D scene with detailed and complex configurations that closely resemble real-world distributions. To this end, we propose ARCHITECT, a generative framework for creating realistic and interactable 3D scenes via diffusion-based 2D inpainting [Podell et al., 2023]. Our pipeline leverages controllable and hierarchical generation in 2D image space. Compared to LLMs which operate in language space, pre-trained image-based generative models are able to better capture scene and object configurations from massive image data readily available, both at the scene level and in fine-grained inter-object spatial information. Pre-trained depth estimation models [Ke et al., 2024, Bhat et al., 2023, Yang et al., 2024a] can then be used to lift the generated 2D static image to 3D environments. However, images created from 2D generative models do not provide accurate camera parameters, which are crucial for reconstructing accurate 3D environments. In addition, the predicted depth images also present scale ambiguity. To address these challenges, we propose to "control" the 2D generative models with 3D constraints via *hierarchical inpainting*. First, we render a photo-realistic image in a simulated empty scene with only a static background, where we have access to ground-truth depth and camera parameters. We then use this image as a *template* for inpainting the foreground using 2D diffusion models. During this process, the generation respects the camera parameters informed by the geometric cues in the background image and ensures that the inpainted components are both semantically and spatially consistent with existing components in the input image. By generating images this way, we effectively control the camera parameters and depth scale for the generated image, which allows us to project it back to 3D point clouds. Subsequently, utilizing visual recognition models [Kirillov et al., 2023, Liu et al., 2023, Ren et al., 2024], we segment the 2D image to obtain the semantics and geometric information of each generated object. These objects are then instantiated in the actual simulated environments, by either retrieving from large-scale asset databases [Deitke et al., 2023a, Mo et al., 2019] or generating using image-to-3D generative models [Xu et al., 2024].

While the pipeline described above is able to generate 3D scene configurations described from a single camera view, our goal is to generate complete scenes observable from *multiple* views. In addition, we aim to generate scenes with real-world complexity, where objects of different scales together form a holistic environment (e.g. ideally we want to also generate small items placed on a shelf or in a drawer). Therefore, we further extend the pipeline to perform iterative and hierarchical inpainting, during which we continuously render new image patches of different locations of the scene to further enhance the complexity when needed. Specifically, given a text description of a target scene, we (i) generate the floor plan following previous works [Yang et al., 2024c, Wen et al., 2023], (ii) add background assets such as walls and floors into a simulated environment, (iii) render images of this empty scene and perform the aforementioned, proposed iterative inpainting process for scene-level generation from multiple camera views, (iv) hierarchically, apply inpainting again at a finer level to place small objects in various semantically plausible locations in the interior space, and (v) finally resulting in a complex 3D scene. Note that such iterative process results in a flexible generative pipeline that can handle different levels of inputs: text descriptions, floor plans, or even pre-arranged scenes.

Our pipeline is able to generate complex scenes that are fully interactable, with detailed asset placement and configurations at multiple scales, as shown in Figure 1. Note that since we make use of powerful prior knowledge encoded in 2D pre-trained generative models, we are able to generate open-vocabulary scenes in a zero-shot manner, for not only diverse room types in home settings, but also non-home environments such as grocery stores. Our experiments show that our framework outperforms prior scene creation approaches in generating interactable scenes that are more complex and realistic. We summarize our main contributions as follows:

- We introduce ARCHITECT, a zero-shot generative pipeline that creates diverse, complex, and realistic 3D interactive scenes to advance Embodied AI agents and Robotics research.

- We propose to leverage 2D prior from vision generative models to facilitate the 3D interactive scene generation process, and make such process *controllable* by initializing from simulation-rendered image for hierarchical inpainting, ensuring consistent spatial features and controllable camera parameters and depth scale, allowing accurate 2D to 3D lifting.

- The experimental results show that our method outperforms previous approaches in generating more complex and realistic interactive 3D scenes, both quantitatively and qualitatively.

| Methods | No Train | No Human Effort | Interactive | Organized Small Objects | Open Vocab |
|---------|:--------:|:---------------:|:-----------:|:-----------------------:|:----------:|
| Behavior-1k | | | ✓ | ✓ | |
| ProcTHOR | ✓ | | ✓ | | |
| Holodeck | ✓ | ✓ | ✓ | | ✓ |
| AnyHome | ✓ | ✓ | | | ✓ |
| RoboGen | ✓ | ✓ | ✓ | | ✓ |
| PhyScene | | ✓ | ✓ | | |
| DiffuScene | | ✓ | | | |
| LayoutGPT | | ✓ | | | |
| Text2Room | ✓ | ✓ | | | ✓ |
| Ours | ✓ | ✓ | ✓ | ✓ | ✓ |

Table 1: We compare our work with previos works that also aims to generate large scale 3D scenes in 5 aspect. Here, **No Train** means no need for training data of indoor layouts.

Our code will be made publicly available.

## 2   Related Works

**Indoor Scene Generation** A large body of works have focused on automatic indoor scene generation. Some works generate only the static mesh of the scene [Höllein et al., 2023, Schult et al., 2023]; in contrast, ours generates interactive scenes that can be used for downstream embodied AI and robotics tasks. One line of research generates interactive indoor scenes via procedure generation with manually defined rules [Deitke et al., 2022]. The quality and diversity of the generated scenes highly depend on the predefined rules, which demands huge human efforts. Another group of works train a generative model (e.g., transformers or diffusion models) on in-door scene datasets [Fu et al., 2021] and use the trained models for scene generation [Yang et al., 2024b, Feng et al., 2024, Tang et al., 2023a, Paschalidou et al., 2021], and the quality and diversity of the scenes are bounded by the training dataset. Ours differ from these two lines of work as we do not use any manually defined rules nor pre-collected datasets, which might constrain the diversity of the generated scenes. Instead, we achieve higher diversity in the generated scenes by leveraging 2D image generative models that are trained with abundant internet data, which cover a wider distribution of scenes than those generated by manually defined rules or a fixed dataset. Recently, several works employ a Large Language Model (LLM) for indoor scene generation [Wang et al., 2023b, Yang et al., 2024c, Wen et al., 2023], such as floor plan, layout, and object placements. Since the 3D spatial reasoning abilities of LLMs are still limited, the quality and diversity of the generated scenes are still bounded. By combining 2D image generative models, simulation rendering and controlled image impainting, our method achieves more coherent 3D layouts and higher scene diversity. We make a comparison between us and previous works in Table 2 .

**Text-to-Image Diffusion models** Text-to-image models based on diffusion model, such as DALL-E2 [Ramesh et al., 2021] and LDM [Rombach et al., 2021], have become dominant in text-to-image generation. These text-to-image diffusion models have been trained on billions of images, giving them strong visual and 3D priors in addition to their image generation capabilities. Through SDS (Score Distillation) loss proposed by DreamFusion [Poole et al., 2022], the 3D prior can be leveraged for a variety of downstream tasks like 3D object generation [Poole et al., 2022, Tang et al., 2023b, Wang et al., 2023c, Liang et al., 2023]. In our observations, in addition to visual and 3D priors, these text-to-image diffusion models also have strong priors for the layout of furniture and objects in a room. Therefore, we designed a pipeline to exploit these layout priors from inpainting diffusion models to generate realistic indoor scenes.

## 3   Method

We automatically generate complex and realistic embodied environment scenes through a hierarchical and iterative process of rendering, inpainting, and visual reasoning. Specifically, as shown in Figure 2, starting from an empty room, our pipeline first iteratively generates the layout of large furniture items (Figure 2 *Large Furniture*). Subsequently, we place smaller objects inside or upon these large furniture pieces, as depicted in Figure 2 *Small Objects*. In detail, our process entails the following

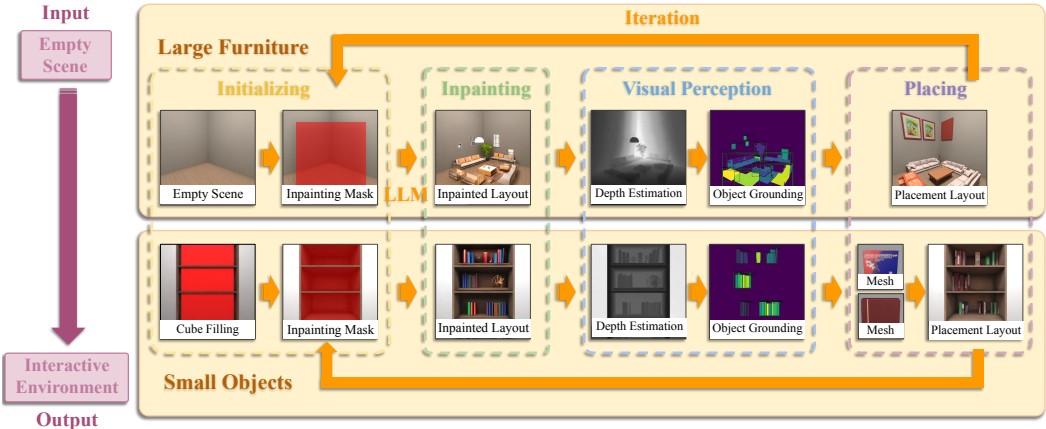

Figure 2: Demonstration of our pipeline that generate complex interactive environment starting from empty scenes, including **Initializing**, **Inpainting**, **Visual Perception** and **Placing** modules.

steps: (1) **Initializing**: We begin by selecting a view in the scene and rendering an image using Pyrender [Matl, 2019] and LuisaRender [Zheng et al., 2022], and generating an inpainting mask; (2) **Inpainting**: Given a text prompt generated by LLMs, along with the rendered image and masks from step 1, we inpaint the image according to the text prompts, leveraging Latent-Diffusion [Rombach et al., 2021]; (3) **Visual Perception**: Upon recognizing the inpainted image, we generate 3D bounding boxes for objects using GPT4v, Grounded-SAM, and Marigold [OpenAI, 2023, Kirillov et al., 2023, Liu et al., 2023, Ren et al., 2024, Ke et al., 2024], as well as the rendering parameters from step 1; (4) **Placing**: We place objects into the scene according to the 3D bounding boxes and return to step 1 to continue generating new objects in the next iteration.

## 3.1 Initializing Module

**Overview** In this module, we initialize the viewpoint for the iteration, rendering an image and obtaining the inpainting mask and ground-truth depth for the following steps.

**Large Furniture** Given an empty room, we heuristically choose the first view that spans from one corner to the opposite corner, maximizing the visible space of the room to render an image, as in Figure 2 *Empty Scene*. Setting a light source in the middle top of the room, we use a ray-tracing based method to render a high-quality image and a raster-based method to obtain the ground-truth depth and object segmentation. Next, we generate an inpaint mask for the image, which is centered within the frame, as shown in Figure 2 *Inpainting Mask*. If objects are already present in the scene, we utilize the object segmentation mask obtained from the rasterizer to filter out those pixels from the inpaint mask, ensuring that any furniture or objects placed within the room will not disappear from the newly inpainted images.

**Small Objects** As illustrated in Figure 2 *Small Objects*, to place small objects on large furniture, we first heuristically choose a front-top view or a front view depending on whether we are placing on top of a large object (e.g., a table) or inside an object (e.g., a shelf). To obtain the inpainting mask, we place a cube within or atop the bounding box of the larger furniture object, with a size slightly smaller than that of the furniture object itself, as shown in Figure 2 *Cube Filling*. In cases placing small objects on top of larger ones, we position the cube on top of the larger object's bounding box, with a fixed height and the other two dimensions slightly smaller than those of the larger object. Similar to large furniture, we remove the pixels of existing small objects from the inpainting mask.

## 3.2 Hierarchical Inpainting Module

**Overview** In this module, we utilize the image and inpainting mask provided from the previous step to first generate text prompts using LLMs. Subsequently, we use these prompts for image inpainting.

The inpainting process for both large furniture and small objects is depicted in Figure 2 *Inpainting*. To ensure smoother inpainting results, we apply techniques such as erosion and Gaussian blur to the mask before commencing the inpainting process. This preparation allows for more effective filling of the contents within the mask. To enhance the diversity and ensure the generation of reasonable objects, we prompt LLMs to automatically generate suitable text prompts and negative prompts for the inpainting model. For example, when we input the configuration of a partially generated living room that includes a TV set into an LLM, the LLM will place the word "TV" in the negative prompt, reasoning that a living room normally has only one TV set.

During this step, we generate multiple inpainted images. If the number of recognized objects in a generated image falls below a predefined criterion, we filter out this image and generate new ones.

### 3.3 Visual Perception Module

**Overview** This module takes the inpainted image, ground-truth depth, and camera parameters to recognize and segment objects, estimate their depth, back-project them into 3D, and finally output 3D bounding boxes for each object.

For object recognition, we initially utilize GPT4v to detect all objects present in the image. The identified object names then serve as tags for Grounding-Dino, which performs object detection and provides the output bounding boxes. Following this, we use the SAM to obtain instance-level segmentation masks based on these bounding boxes, as depicted in Figure 2 *Object Grounding*.

After that, we estimate the relative depth of the generated image and rescale the predicted depth using reference depth. Specifically, giving $n$ reference pixel set $P_r = \{(i_0, j_0), ..., (i_{n-1}, j_{n-1})\}$, referenced depth map $D_r$ in $R^{W \times H}$ where $W$ and $H$ are the resolution of the image, estimated depth map $D_e$ in $R^{W \times H}$, we rescale the estimated depth map to $D_{rescaled}$ in the following formulas:

$$\max_r = \max_{t \in [0,n-1]} \left(D_r^{(i_t,j_t)}\right), \min_r = \min_{t \in [0,n-1]} \left(D_r^{(i_t,j_t)}\right), \max_e = \max_{t \in [0,n-1]} \left(D_e^{(i_t,j_t)}\right)$$

$$\min_e = \min_{t \in [0,n-1]} \left(D_e^{(i_t,j_t)}\right), \text{scale} = \frac{\max_r - \min_r}{\max_e - \min_e}$$

$$D_{rescaled} = D_e \cdot \text{scale} - \frac{1}{n} \sum_{t \in [0,n-1]} D_e^{(i_t,j_t)} \cdot \text{scale} + \frac{1}{n} \sum_{t \in [0,n-1]} D_r^{(i_t,j_t)}$$

We employ different strategies when selecting reference pixels $P_r$. For placing large furniture, we utilize all the non-masked parts of the image as reference pixels to ensure general consistency with the room's floors and walls. For small objects, we focus on the non-masked areas of large furniture as reference pixels, aiming for consistency specifically with the inpainted object. This approach is adopted because the depth information outside the inpainted objects exhibits discontinuities that are challenging to predict accurately. For instance, as shown in Figure 2 *Depth Estimation*, predicting the depth of the wall behind the shelf is difficult and could introduce noise if considered.

Once the depth estimation is acquired, we use the camera parameters from the rendering process to back-project the depth into a 3D point cloud. Utilizing the 2D masks provided by SAM, we extract the point cloud for each object instance. To eliminate any outliers, we apply DBSCAN[Khan et al., 2014] clustering to each segmented object, which allows us to derive axis-aligned bounding boxes for each object.

### 3.4 Placing Module

**Overview** Equipped with the 3D bounding boxes of objects, this module places the objects into the simulation and returns to the **Initializing** phase to commence the next iteration.

**Large Furniture** To incorporate large furniture into a scene utilizing 3D bounding boxes, we initially retrieve each piece of furniture according to a text description generated by GPT4v. After extracting a list of instances from the datasets, we select them based on feature similarity and the proportionality of their scale in three dimensions. The scale of each item is adjudicated by large language models using common sense knowledge.

Owing to the possibility that the retrieved furniture may not precisely conform to the 3D bounding boxes and minor errors in depth estimation, directly placing furniture at the center of the bounding

box can lead to issues such as collisions or complications arising from partial view observations. To mitigate these issues, we adopt an alternative approach by generating constraints derived from the 3D bounding boxes. We employ search methods akin to those used in Holodeck [Yang et al., 2024c] to determine optimal placement. Unlike Holodeck, which utilizes LLMs to generate all constraints, we derive ours directly from the generated images and 3D bounding boxes. This search process enables us to avoid collision conflicts while simultaneously ensuring alignment with the generated image. Further details about these constraints can be found in Appendix A.

**Small Objects** To position small objects within a scene as defined by 3D bounding boxes, we first generate 3D instances based on the semantic content of each object and then place them at the specified positions within the 3D bounding boxes. Subsequently, we adjust the orientation and scale of the small objects to match the orientation and size of the bounding boxes. Notably, due to the partial nature of point clouds, it is impractical to uniformly apply the scale of the bounding box across all three dimensions. Instead, we focus on utilizing the scale along the dimensions perpendicular to the viewing direction. The placement process for small objects is depicted in Figure 2.

## 4 Experiments

**Dataset** We retrieve objects from Objaverse. Deitke et al. [2023b,a] Objaverse is a dataset that contains massive annotated 3D objects. It includes various objects, including manually designed objects, everyday items, historical and antique items, *etc*. In the process of generating indoor scene objects, we retrieve suitable furniture from the Objaverse dataset and place them in the scene.
We also retrieve articulated objects from PartnetMobility [Xiang et al., 2020]. PartnetMobility contains 2346 3D articulated models from 46 categories, with articulation annotations.

**Implementation** We use Marigold [Ke et al., 2024] as the depth estimation model. We use Grounded-Segment-Anything as our segementation model. We use the SD-XL [Podell et al., 2023] inpainting model provided by diffusers as the image inpainting diffusion model. We use LuisaRender [Zheng et al., 2022] as our renderer. For text-to-3D generation, we first use MVdream [Shi et al., 2023] to generate a image and then feed the image to InstantMesh [Xu et al., 2024] to generate the 3D asset. All experiments, including qualitative evaluation, quantitative evaluation and robotics task are all conducted on an A100 GPU.

**Baseline** We compare ARCHITECT with state-of-the-art indoor scene generation approaches: (1) Holodeck [Yang et al., 2024c], leveraging common sense from LLMs to generate floor plans and place objects; (2) Text2Room [Höllein et al., 2023], utilizing the knowledge from image diffusion models and depth estimation models to generate the entire mesh of a scene; (3) DiffuScene [Tang et al., 2023a], learning a diffusion model to generate the layout of 3D objects in a scene. In addition to the methods mentioned above, AnyHome [Wen et al., 2023] is another baseline we would like to compare with. However, the method is not yet fully open-sourced, so we will leave it for future work.

**Metric** To evaluate the semantic correctness of the generated scenes, we use the following metrics that calculate the similarity between a rendered image and a given caption: (1) CLIPScore [Hessel et al., 2021], computing the correlation between image feature of the rendered image extracted by CLIP image encoder and text feature of the caption extracted by CLIP text encoder; (2) BLIPScore, using the image-text-matching head of BLIPv2 [Li et al., 2023b] to compute alignment between the rendered image and caption; (3) VQAScore [Lin et al., 2024], feeding the rendered image and caption into a VQA model, returning the probability that the answer to the question "Does the image show caption" is "Yes" as the score; (4) GPT4o Ranking, asking GPT4o to rank rendered images of generated scenes, and calculate the average ranking as the score. We also conduct a user study to evaluate various aspects of the generated scenes. We ask users to rate the following four indicators from 1 to 5: Visual Quality (VQ), Semantic Correctness (SC), Layout Correctness (LC) and Overall Preference (OP).

### 4.1 Scene Generation

**Qualitative Evaluation** We compare rooms generated by our model with those generated by other methods. Figure 3 shows the results for both household and non-household scenes. The comparisons with Text2Room and DiffusScene are provided in Appendix B, as these methods generate either non-interactive scenes or lack diversity.

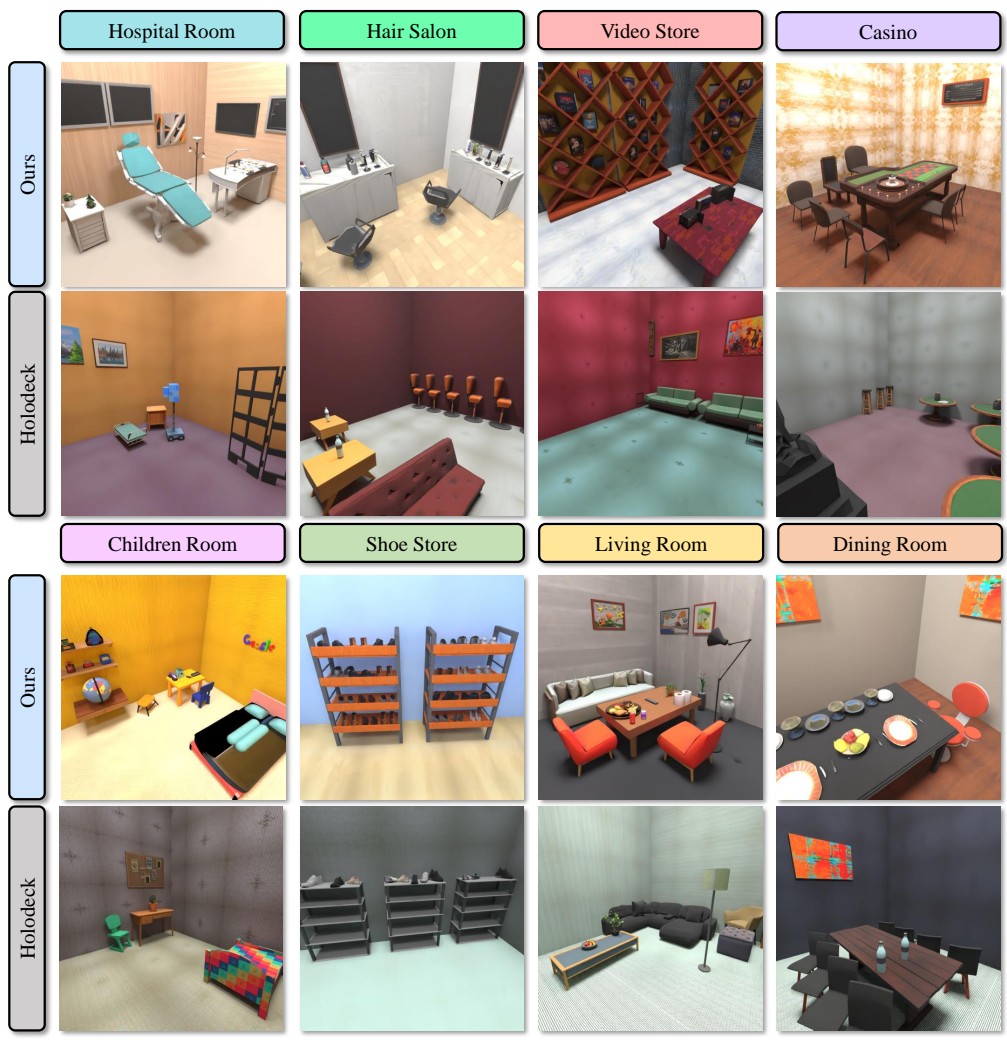

Figure 3: We compare ARCHITECT with other methods in both household scenes(living room and dining room) and other non-household scenes. We only compared the household scene generated by Diffuscene due to its limitations in Figure 7 and compared with Text2Room in Figure 8.

Compared to Holodeck, our scenes are more realistic, leveraging the capabilities of 2D diffusion models. For example, in the hair salon scenario, Holodeck fails to generate semantically correct scenes because the spatial constraints and objects are entirely generated by LLMs. Additionally, our work demonstrates the ability to generate more complex and detailed placements of small objects. For instance, the shoe store filled with paired shoes, toys on the children's room shelf, and the organized placement of items on the dining room table all exceed the generative abilities of Holodeck. This comparison highlights the effectiveness of our iterative and hierarchical inpainting process, showing that the use of 2D generative models indeed brings more spatial priors compared to LLMs.

**Quantitative Evaluation** We compare ARCHITECT to other state-of-the-art indoor scene generation results using 2D image scores and user studies. The results are shown in Table 2 and Table 4. ARCHITECT outperforms others in CLIP score, BLIP score, and GPT-4o ranking, while achieving a relatively high VQA score (only slightly lower than Text2Room). It demonstrate that our generated scene are generally better aligned with the room caption (with better semantic and layout coherence). In GPT-4o's explanations of it's ranking, we found some common points of our previous analysis. Rooms generated by Text2Room are often criticized for having artifacts and distortion and as a result are often ranked lower; rooms generated by Holodeck are often described as simply

| Method | Text-Image Scores | | | | User Study | | | |
|---|---|---|---|---|---|---|---|---|
| | CLIP↑ | BLIP↑ | VQA↑ | GPT4o↓ | VQ↑ | SC↑ | LC↑ | OP↑ |
| Diffuscene | 0.6785 | 0.4310 | 0.7561 | - | 3.76 | 3.52 | 3.37 | 3.50 |
| Text2Room | 0.6491 | 0.1223 | **0.8149** | 2.64 | 2.79 | 3.62 | 3.04 | 3.12 |
| Holodeck | 0.6502 | 0.3463 | 0.5696 | 1.91 | 3.34 | 3.14 | 3.11 | 3.07 |
| Ours | **0.7173** | **0.5859** | 0.8073 | **1.36** | **3.87** | **3.76** | **3.65** | **3.71** |

Table 2: Quantitative Comparison. We evaluate 2D image metrics, including CLIP Score, BLIP Score, VQA Score and GPT4o ranking. We also conducted a user study, reporting visual quality(VQ), semantic correctness(SC), layout coherence(LC) and overall preference(OP). GPT-4 ranking involves ranking and therefore does not include Diffuscene which can only generate a limited number of household scenes.

arranged; rooms generated by ARCHITECT are more favored by GPT-4o evaluator. In user study, ARCHITECT outscored other methods in all four aspects. It is worth noting that the visual quality score of Text2Room is significantly lower than other methods, which is likely due to the artifacts and distortions in Text2Room-generated scenes.

## 4.2 Embodied/Robotic task

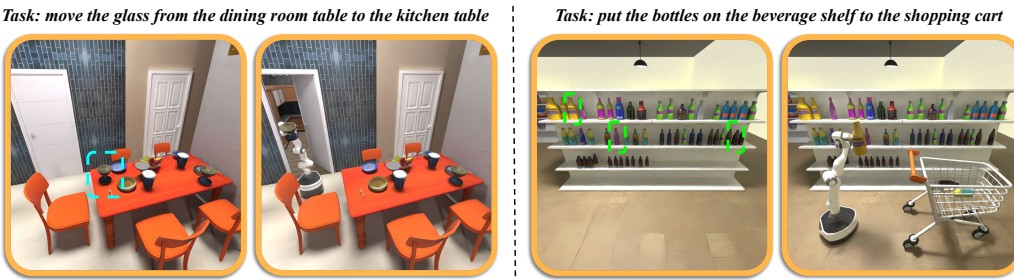

Figure 4: Two robot manipulation tasks generated in our scene setting.

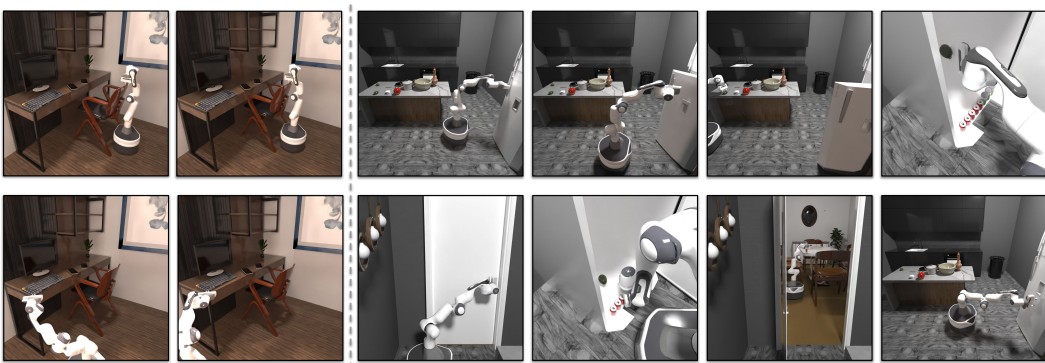

Figure 5: **Left:** the robot organizes the room by pushing the chair under the table and pushing the keyboard inside the table. **Right:** the robot opens the fridge door, grasps the mango and puts it into the fridge, opens the kitchen-dining room door, grasps the soda can and puts it on the dining room table, and finally closes the fridge.

Inspired by previous work RoboGen [Wang et al., 2023b], we are making efforts to collect large-scale data for long-term embodied or robotics tasks in our generated scene.

A significant challenge we face is that the inclusion of all the detailed small objects in the simulation significantly slows down the speed of the embodied environment. To address this, after generating the scene, task and task decomposition, we use LLMs to select relevant objects for each substep, while all other objects are designated as background objects and will not be physically simulated during this substep.

Given our house-level scene generation pipeline, we can now extend RoboGen to generate action trajectories for skills that require long-distance navigation and more complex tasks. Specifically, by inputting the floor plan, large furniture, and small objects into GPT-4, it first generates a task related to the existing objects in the scene. Then, as in RoboGen, it decomposes the task and filters out irrelevant objects to the background. Finally, we leverage action primitives and training supervision generated by LLMs to obtain a trajectory of actions to solve the task. We demonstrate two example generated tasks in Figure 4, and two other task with corresponding collected trajectories in Figure 5. The comparison of diversity of is shown in Table 3 by the self-BLEU score of task discription.

### 4.3 Object Generation

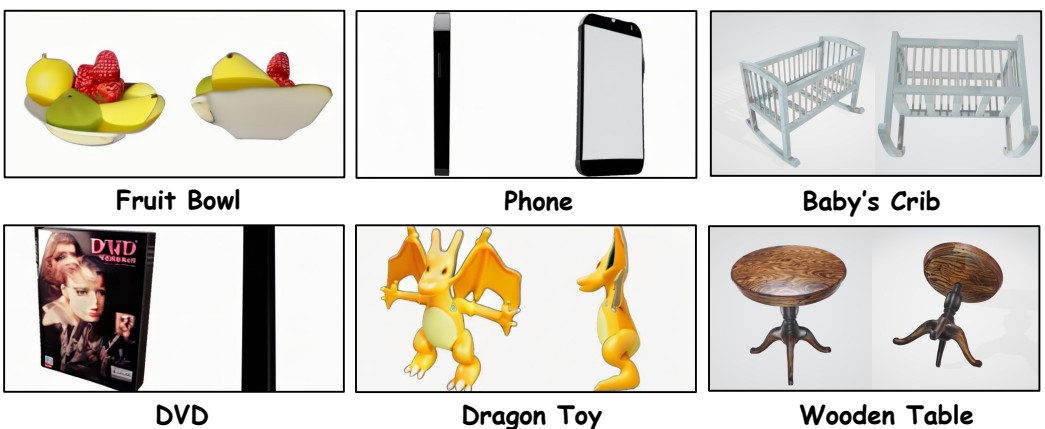

Figure 6: Examples of generated small objects and large furniture.

To address one of our limitations, the dependence on a large furniture database, we apply a pipeline to generate high-quality large furniture. It optimizes a differentiable tetrahedron mesh with SDS loss [Guo et al., 2024], using the normal-depth diffusion model and albedo diffusion model provided by RichDreamer [Qiu et al., 2024] as the main supervision signal. This pipeline is capable of generating high-quality object meshes from text guidance, specifically large furniture in our case. Some results are shown in the right part of Figure 6 right part. We also show some qualitative resutls about small object generation in Figure 6 left part, which is another crucial factor of the quality of generated scenes.

## 5 Conclusion and Future Work

In this paper, we propose ARCHITECT, a generative framework capable of creating *diverse*, *realistic*, and *complex* Embodied AI environments. Leveraging pre-trained 2D image inpainting diffusion models that better capture scene and object configurations compared to LLMs, ARCHITECT iteratively extracts diverse and realistic layouts from image inpainting results. We also propose to *control* this inpainting process by processing geometric cues in the background of a rendered image. This process effectively controls the camera parameters and depth scale for the generated image, allowing us to project it back into 3D point clouds. The scenes generated by ARCHITECT provide realistic and complex environments for downstream Embodied AI and robotics applications. In qualitative and quantitative comparisons, ARCHITECT outperformed baseline methods in both realism and diversity. We believe ARCHITECT is an important step towards creating large-scale interactive 3D environments.

**Limitation and Future work**   Currently, ARCHITECT retrieves furniture and large objects from datasets. This means that the diversity of furniture in our results is inherently limited by the dataset. In the future, we will explore generative methods to create more high-quality and articulated objects to further enhance the diversity of the generated scenes.

## Acknowledgement

We thank the anonymous reviewers for their helpful suggestions. This work is funded in part by grants from Microsoft Accelarate Foundational Models Research Initiative, Cisco, and NSF IIS-240438.

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

# 6 Appendix

# A Implementation Details

## A.1 Constraint and Search

Furniture can be devided into floor objects and wall objects. In detail, for the floor objects, we rely on the following type of constraints for furniture: *Global*, *Location*, *Distance*, *Relation*, *Alignment* and *Rotation*.

/* Global Constraint */
**Edge:** at the edge of the room, close to the wall.
**Middle:** not close to the edge of the room.
**Corner:** at the corner of the room.
**Horizontal/Vertical:** the global direction.

/* Distance Constraint */
**Near, object:** near to the other object.
**Far, object:** far away from the other object.

/* Relation Constraint */
**In front of, object:** in front of another object.
**Behind, object:** behind of another object object.
**Left of, object:** to the left of another object.
**Right of, object:** to the right of another object.

/* Alignment Constraint */
**Center aligned, object:** aligned with another object .

/* Soft Location Constraint */
**Location, (x, y):** predicted bounding-box location.

/* Rotation Constraint */
**Face to, object:** face to the center of another object.

Different from Holodeck that is using LLMs to generate all the constraints, all above constraints are generated by sorting floor objects by size and traversing them based on the position of their bounding boxes except for the rotation constraints. Specifically, for each floor object's bounding box, constraints of each type are assigned based on the distance and directional relationship between its boundaries and those of other floor object bounding boxes. In particular, rotation constraints cannot be solely determined by the bounding box, so an LLM is consulted to obtain the rotation constraint based on common sense (e.g., chairs facing a table). After obtaining the complete constraints, a DFS algorithm is utilized to explore possible placements for each item. Placements that do not meet the hard constraints are filtered out, and the highest scoring placements are selected based on the soft constraint scores. Here, hard constraints refer to mandatory constraints, such as global constraints and position constraints. Soft constraints refer to cumulative scores, where the highest scoring options are prioritized, such as location constraints. In practice, we applied a greedy pruning mechanism to the DFS algorithm, exploring only 3 nodes with highest score at each time. The score $S_p$ for each placement is calculated as follows:

$$S_p = W_{loc} \cdot \left( \sum_{i \neq i} \Delta_i \cdot w_i + \frac{w_{cur}}{\Delta_{cur}} + C \right) + W_{rotation} \cdot \left( \sum_{i=1}^{n} \mathbb{1}_{r_i} \right)$$

For each current object, $W$ represents the weight of constraints; $w$ represents the weight of each object; $\Delta$ represents the deviation from the reference, which hopes to be close to the reference and away from other items already placed; $C$ represents constants to keep result positive; $\mathbb{1}$ is a indicator function that equals 1 if the rotation satisfies the constraint; $r$ represents the rotation of the item.

As for wall objects, most of its constraint comes from floor. Wall objects constraints are as follows:

/* Global Constraint */
**Above, object:** close to the wall, above a specific floor object.

/* Soft Location Constraint */
**Location, (x, y):** predicted bounding-box location.

/* Position Constraint */
**Height:** The height of the object.

The placement of wall objects is relatively simpler because it does not require specifying an orientation; by default, they face away from the wall. Additionally, the likelihood of conflicts on the wall is lower, and using soft location constraint suffices for effective arrangement. And the searching process of wall objects is just almost the same as floor objects.

## A.2 Large Furniture Retrieving

Following Holodeck, for each piece of large furniture, we first retrieve multiple candidates from the dataset using text descriptions of the assets. Then, we select one asset from the retrieved candidates based on scale similarity, which is calculated as the L1 difference between the scale of 3D bounding box of object point cloud and the 3D bounding box of object mesh. Additionally, we integrated image similarity using the cosine similarity of CLIP features in the selection process in our latest pipeline. Here, scale similarity and image similarity are used only in the candidate selection process rather than the retrieval process, since there could be significant occlusions in the image (e.g., a chair behind a table) that could greatly influence the accuracy of retrieving.

## A.3 Small Objects Generation and Selection

We use a text-to-3D pipeline (text-to-image and image-to-3D) to generate 3D assets for small objects. To make the scene more reasonable and resemble the inpainted image, we generate multiple candidates for each type of object and use the cosine similarity of DINO features to select from the candidates. We also experimented with an image-to-3D pipeline, starting from the object image segmented from the inpainted image. However, the resolution of the segmented image is low, resulting in sub-optimal 3D shapes and textures.

## A.4 View Selection

For large furniture placement, we heuristically select up to three views (right-back corner to left-front corner, front middle to back middle, and left-back corner to right-front corner) that can cover the whole room area for inpainting. Assuming the room ranges from $(0, 0)$ to $(x, y)$, the three views would be looking from $(x, y, 1.8)$ to $(0, 0, 0.5)$, from $(\frac{x}{2}, 0, 1.8)$ to $(\frac{x}{2}, y, 0.5)$, and from $(0, y, 1.8)$ to $(x, 0, 0.5)$. We stop inpainting from new views when the occupancy of the room is larger than 0.7 or it has been inpainted from all three views.

Additionally, we use an 84-degree FOV for our camera during rendering, a standard parameter for real-world cameras. Consequently, for a square room, this setup results in approximately 95 percent of the room being visible from a single corner-to-corner view.

For small object placement, we first ask LLMs to determine which objects can accommodate small objects on or in them, and then inpaint each of them with heuristic relative views. For objects like tables or desks on which we are placing items, we use a top-down view. For shelves or cabinets in which we are placing objects, we use a front view. The distance of the camera from the object is adjusted according to the scale of the object and the camera's FOV, ensuring the full object is visible during inpainting.

# B  More Experiments

## B.1  More Comparison Cases

As shown in Figure 7 and 8, compared to Text2Room, our method generates more photorealistic scenes. However, since Text2Room directly projects RGB pixels into 3D space using depth maps, artifacts are unavoidable, the geometry of the generated 3D mesh is distorted, and it can't serve as an embodied environment since it's not interactive. These issues are addressed in retrieval-based methods. Compared to Diffuscene in Figure 7, our method generates more reasonable and detailed scenes. Our results exhibit more detail and precision. Additionally, the dataset lacks information on the placement of small objects, limiting the diversity of small object placement in Diffuscene-generated scenes.

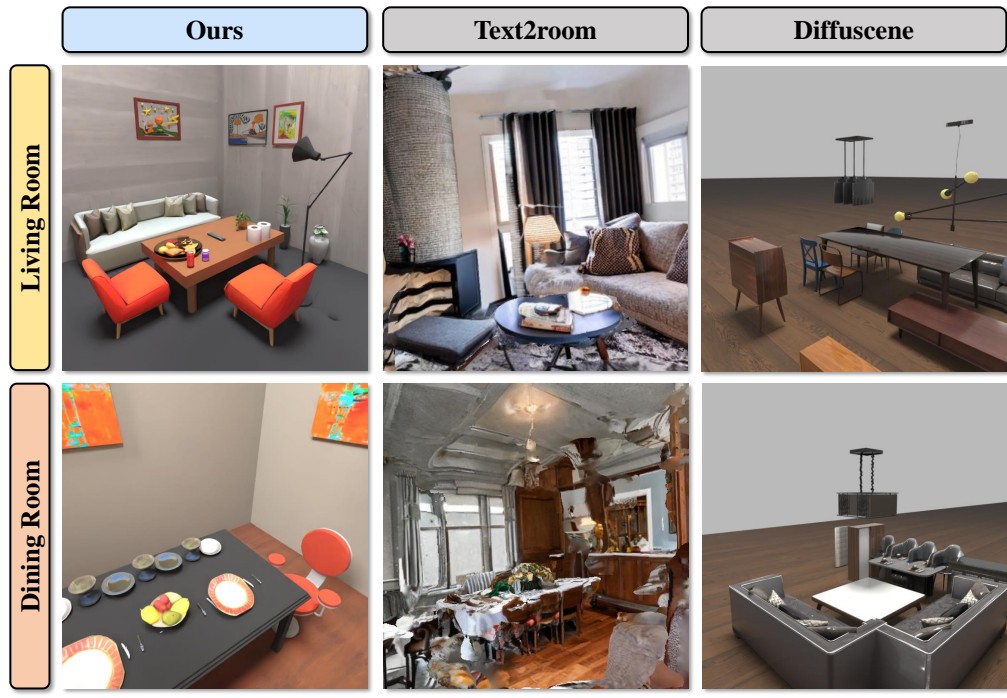

Figure 7: Comparison of living room and dining room scene generated by ARCHITECT, Text2room, Diffscene.

## B.2  Comparison with Baselines

The comparison with PhyScene is shown Table 4, where we present the comparison results for generated living rooms (only the weight of living room generation is released).

In our experiments, we aim to provide a general comparison with three types of related works: works that generate only the static mesh, works that are trained on existing datasets and works that generate open-vocabulary scenes using foundation models.

It's challenging to make a completely fair comparison between our methods and the Text2Room method since they serve different purposes. While Text2Room generates the entire mesh without retrieving objects, none of its assets are interactive and it may achieve higher photorealism by directly generating meshes from 2D images.

## B.3  Controllability and Editing

In short, our method combines a diffusion-based pipeline with an LLM-based method, which still possess the ability of controlling and editing. The inpainting-to-layout pipeline functions

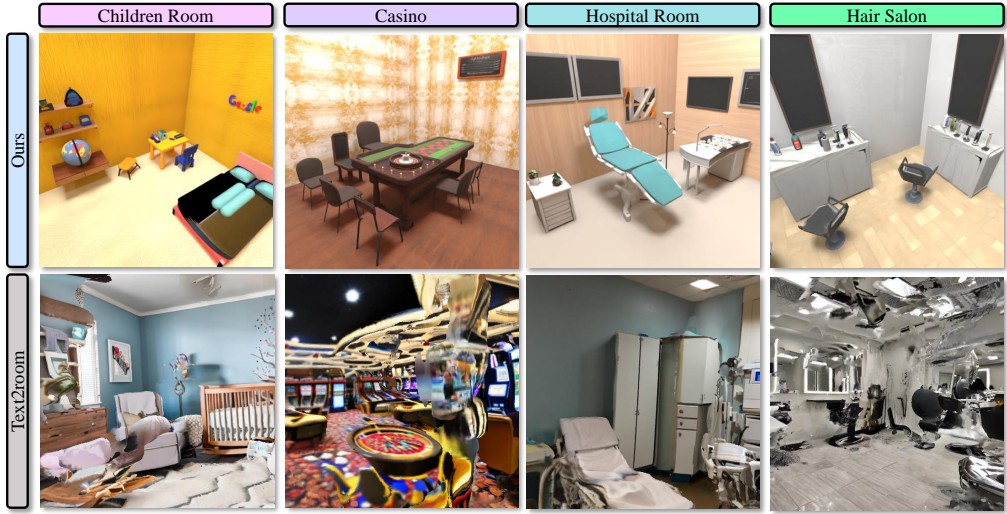

Figure 8: Comparison of four scenes generated by ARCHITECT, Text2room.

can be considered as an API function callable by the LLM. Our approach aims to generate scene configurations seeded from diffusion models, with scene editing as an othogonal feature enabled by LLMs. Specifically, the scene configuration generated by our pipeline can be represented by each object's name, position, scale, bounding box, orientation, and asset UID, which can be easily converted to text representations. This allows us to feed this information directly into LLMs to further control or edit the scene layout. Corresponding experiments could be found in Appendix C.

### B.4 Comparison of Architect and LLMs in Small Object Placement

It's challenging for LLMs to directly solve arrangement problems. First, for small object placement on shelves, LLMs lack information about supporting surfaces, making it impossible for them to solve this issue. Second, for placement on tables, while we might know the supporting surfaces given the bounding boxes, LLMs struggle with object orientations, often resulting in less complex scenes or scenes with severe collisions. We show a comparison of small objects generated by our methods and LLMs in the middle part of Figure 9 and in Table 3.

### B.5 Similarity of Generated Scenes and Inpainted Images

We've also evaluated the image similarity between inpainted images and images of generated scenes against empty scenes, as shown in Table 3. The results indicate that, although not exactly the same, the generated scenes are to some degree faithful to the image generation results.

### B.6 Consistency of Inpainting

The appearance of the masked area is consistent with other areas both stylistically and geometrically. We also apply a commonly used technique, softening the boundary of inpainting masks, to improve consistency. A comparison of the results before and after using softened inpainting masks is shown in the left part of Figure 9.

|  | Large Furniture | Small Objects |
|---|---|---|
|  | Similarity (%) ↑ | Similarity (%) ↑ |
| Empty vs. Inpaint | 45.33 | 77.85 |
| Inpaint vs. Placed | **85.04** | **83.83** |
| LLM Placement VQScore ↑ | 74.93 | |
| Our Placement VQScore ↑ | **80.81** | |
| RoboGen Self-BLEU ↓ | 0.284 | |
| Ours Self-BLEU ↓ | **0.198** | |

Table 3: Quantitative experimental Results.

| Method | CLIP ↑ | BLIP ↑ | VQScore ↑ |
|---|---|---|---|
| PhyScene | 71.42 | 46.51 | 88.72 |
| Holodeck | 69.37 | 53.23 | 84.06 |
| Text2Room | 64.91 | 12.22 | 90.73 |
| Diffuscene | 65.32 | 49.95 | 87.28 |
| ARCHITECT (Ours) | **72.96** | **63.62** | **94.58** |

Table 4: Experimental result of comparison with PhyScene.

## C   Scene Editing

To demonstrate that our pipeline is compatible with scene editing and complex text control, we implemented additional APIs to add, remove, and rescale objects, enabling LLMs to edit the scene.

Initial results for scene editing are shown in the right part of Figure 9. We issued commands to LLMs such as replace the books on the shelf with vases, replace the bookshelf with a cabinet, and make the bookshelf smaller. The LLMs achieved the correct results by calling the provided APIs.

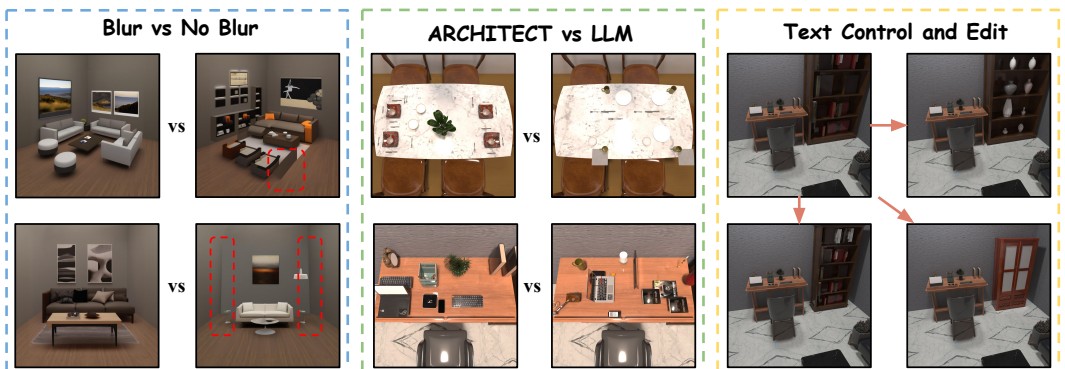

Figure 9: Demonstration of comparison between different mask, different small objects placement and effect of text control. Red dashed box indicates inconsistency when using non-blur mask.

## D   User Study Details

We conducted comprehensive human evaluations to assess the quality of ARCHITECT scenes, with a total of 115 undergraduate students and graduate students participating in the user studies. All participants were volunteers without compensation.

We first provided participants with two minutes to read the instructions, the specific content of which was as follows:

· Thank you for your participation! This questionnaire is used for the experimental part of scientific research articles, which requires human evaluation. We will keep the information of the participants confidential.
· The estimated total time is about 10 minutes.
· There are some pictures in the questionnaire from different utils. Just observe and score all of these pictures. The higher the score, the better the quality.
· An example is shown in the following image:

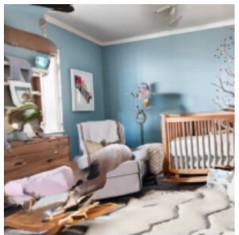

\* 1. Please rate the following scene image：
scene name：children room

|  | 1 | 2 | 3 | 4 | 5 |
|---|---|---|---|---|---|
| visual quality | ○ | ○ | ○ | ○ | ○ |
| semantic correctness | ○ | ○ | ○ | ○ | ○ |
| layout coherence | ○ | ○ | ○ | ○ | ○ |
| overall preference | ○ | ○ | ○ | ○ | ○ |

Figure 10: The example questionnaire for participants.

Then, we randomly assigned each volunteer 23 scenes and asked them to score the scenes from one to five, considering visual quality, semantic correctness, layout coherence, and overall preference. Each volunteer received only 23 scenes to ensure they could complete their responses in approximately 10 minutes.

We collected ratings from all participants and calculated the average scores for these four metrics. These average scores were used to evaluate our model, Holodeck, Text2Room and DiffuScene allowing us to compare the scene generation performance of our model with the two baseline models.

The average response time was 525 seconds, with the longest response time being 1113 seconds and the shortest being 150 seconds. Responses with a duration of less than 230 seconds were filtered out.

# E  Prompts

## E.1  Object Recognition Prompt

> Detect all objects in the picture, generate a description for each object, and decide whether it is floor-object or wall-object.
>
> Here are the definitions of object types:
> floor object: object that is placed on floor or in direct contact with the floor.
> wall object: object that is placed on wall and not in contact with the floor.
>
> Here is a sample answer:
> table: A big yellow table | floor-object
> chair: A gray armchair | floor-object
> tv: A black wall-mounted television | wall-object
>
> Requirements:
> 1. Description should not be too long.
> 2. You should only give the result and no unnecessary words.
> 3. Don't describe the positional relationship between objects.
> 4. Classification can only be **floor-object**, **wall-object**.
> 5. Please pay attention to only large furniture like sofa, table, lamp, shelf, and ignore small objects like bottles or books.

The prompt above is fed into GPT-4V along with an image generated by a 2D inpainting model. The prompt asks GPT-4V to recognize all objects in the inpainting image, briefly describe them, and then classify them as either objects on the floor or objects on the wall. The generated object names will be used to prompt Grounded-SAM. The generated descriptions will be used for retrieving objects or for text-to-3D generation.

## E.2  Inpainting Prompt and Generation Prompt

> Given the objects in the current scene, please list which objects have already reached their potential limits, and the objects are still lacking.
>
> Your answer should be in the following format:
> reached limit: object A, object B, ...
> lacking: object C, object D, ...
>
> The objects in the current scene are: /* a list of objects with quantities, eg: 2 sofa, 1 coffee table, 1 TV*/
>
> Remember, do not answer anything not asked. The lacking objects should ideally contain objects that are not in the scene. The lacking objects you list should be precise, do not give things like "other furniture".

The prompt above asks GPT-4V to provide negative prompts and positive prompts in addition to the room caption for the inpainting model. ROOM-CAPTION will be substituted with the actual room caption. The lacking objects will be added to the positive prompt, and the objects that have reached their limit will be added to the negative prompt.

# F  Societal Impacts

## F.1  Positive Impacts

- **Advancements in Robotics and AI**: ARCHITECT enhances the development of versatile robots capable of assisting in various tasks.

- **Educational Tools**: Generated 3D environments can be used for immersive learning experiences, aiding in the understanding of spatial relationships and complex systems.
- **Accessibility**: Improved AI environments can lead to the development of assistive technologies for individuals with disabilities, enhancing their quality of life.

## F.2  Negative Impacts

- **Job Displacement**: Advanced AI and robotics could potentially displace jobs in certain sectors, necessitating consideration of economic and societal impacts.
- **Bias and Fairness**: Ensuring training data and algorithms are representative and fair is crucial to avoid perpetuating existing biases.
- **Misuse of Technology**: Inferring internal geometric structures of 3D objects could be misused for unauthorized reproduction or surveillance, leading to ethical and legal concerns.

