# OpenReview forum: "Architect: Generating Vivid and Interactive 3D Scenes with Hierarchical 2D Inpainting"
_NeurIPS.cc/2024/Conference — NeurIPS 2024 poster_

### Official Review · Reviewer_4D8M · 2024-07-11

**Soundness:** 3
**Presentation:** 2
**Contribution:** 2
**Rating:** 5
**Confidence:** 4

**Summary:**

The paper proposed a pipeline for 3D scenes generation, which leveraging 2D prior from  diffusion-based image and depth generation. The main proposed point is the hierarchical inpainting, a inpainting mask is generated from a simulated 3D environment, providing a good condition to control the 2D image generation.

**Strengths:**

1. The paper demonstrates a possible approach towards 3D scenes generation by leveraging the prior from 2D diffusion models.
2. The proposed method can support the small objects, which is an advantage compared to other methods.

**Weaknesses:**

1. The effectiveness of the proposed hierarchical inpainting is not well verified, without the control of *3D constraints, how is the quality of the generated scenes?*
2. In 4.2, the paper presents a quite simple example for embodied tasks, this is an important application scenario for 3D scene generation, more results are required to verify the effectiveness of the proposed methods can be used for embodied tasks.
3. typo  in line 16: hierachical → hierarchical

**Questions:**

1. The operation of the depth map is not clear, what is the meaning of *estimated depth map and referenced depth map,  what is the goal of the scaling?
2.   How many iterations is needed? and what is the ending condition?
3. the showed results are conditioned simple texts, does the method supports more complex text descriptions?

**Limitations:**

I don't see any negative societal impact.

---

> ### Author Rebuttal · Authors · 2024-08-06
>
> # Response to Reviewer 4D8M
> *Thank you for your insightful and constructive comments! We discuss some of your questions and concerns below.*
>
> **1. Without the control of 3D constraints, how is the quality of the generated scenes**
>
> To clarify, **the 3D constraint is generated by the hierarchical inpainting** (refer to lines 232 to 245 in the paper). Specifically, after acquiring estimated 3D bounding boxes, if we directly place furniture according to them, it would be less accurate since the point cloud is partial. As a result, we first generate constraints based on those bounding boxes, such as *A is around (x, y, z)* or *A is on the left of B*. We then solve these constraints while enforcing collision-free placement between objects to obtain the final result that best satisfies the constraints.
>
>
> **2. More results for embodied tasks**
>
> We show more experimental results in General Response 2A.
>
> **3. Typo**
>
> Thank you for mentioning! We will fix that.
>
> ### Questions
>
> > **Q1: The operation of the depth map is not clear, what is the meaning of estimated depth map and referenced depth map, what is the goal of the scaling?**
>
> In the pipeline, we first render an image and then inpaint within this image.
>
> For the rendered image, we have the ground truth depth for each pixel since it is rendered in simulation, which serves as the reference depth map.
>
> For the inpainted image, we estimate per-pixel depth using Marigold, resulting in the estimated depth map.
>
> Since Marigold[1] generates only a normalized depth map (and other monocular depth estimation methods might also be confused by the scale of depth), we align the scale of the estimated depth with the reference depth. This alignment is based on the fact that the depth values should match in the uninpainted pixels.
>
> > **Q2: How many iterations is needed? and what is the ending condition?**
>
> As mentioned in General Response 3C, we select up to three views for large object placement, and one view to place small objects for each selected large object. The process stops after these views have been used. The iterations needed for placing large furniture in one room are up to 3, and the iterations needed for placing small objects are conditioned on the number of large furniture pieces present.
>
> > **Q3: The showed results are conditioned simple texts, does the method supports more complex text descriptions?**
>
> It does support more complex text descriptions, benefiting from the hierarchical structure. While diffusion models alone can't handle complex prompts, we use an LLM as central control to interpret complex descriptions and distribute tasks to the diffusion models. For example, given a text description like *a living room with a shelf full of books and a desk with a laptop on it*, the LLM will include the shelf and desk in the positive prompt of the diffusion model to ensure their presence. It will also adjust the prompt to *a shelf full of books* to place books on the shelf and include the laptop as a positive prompt when placing objects on the desk.
>
> A similar idea is demonstrated in General Response 2B.
>
> [1] Ke, Bingxin, et al. "Repurposing diffusion-based image generators for monocular depth estimation." Proceedings of the IEEE/CVF Conference on Computer Vision and Pattern Recognition. 2024.
>
> *We wish that our response has addressed your concerns, and turns your assessment to the positive side. If you have any more questions, please feel free to let us know during the rebuttal window.*
>
> Best,
>
> Authors

---

> > ### Comment · Reviewer_4D8M · 2024-08-12
> >
> > Thanks for the response, and most of my concerns are addressed. I suggest the authors to put some important details into main paper. I have raised my score.

---

> > > ### Author Response · Authors · 2024-08-13
> > >
> > > We are pleased to hear that your evaluation has shifted positively and that your concerns have been addressed. We plan to incorporate these details into the main paper in future versions. If there’s anything else we can do to further elevate your opinion of our work, we would be happy to provide additional information. Thank you once again!

---

### Official Review · Reviewer_Te9g · 2024-07-12

**Soundness:** 3
**Presentation:** 4
**Contribution:** 2
**Rating:** 5
**Confidence:** 4

**Summary:**

This paper aims to generate interactive 3D scenes suitable for downstream tasks like robotics. The paper first generates an empty scene and then utilizes pre-trained 2D inpainting models to fill in the ‘foreground’ and apply visual recognition and depth estimation models to ‘lift’ the 2D objects to 3D space via retrieval or off-the-shelf image-to-3D models. Finally, to generate complete and complex scenes observable from multiple views, this paper further applies iterative and hierarchical inpainting. In summary, the framework can generate complex and interactive scenes with detailed asset placement and shows superior results compared with previous works in complexity and realism.

**Strengths:**

1. The paper successfully mixes a bunch of off-the-shelf base models including GPT4v, Grounding-DINO, SAM, MDE(monocular depth estimation), and inpainting models as well as a bunch of techniques like depth alignment to generate 2D content and then lift them into 3D.
2. The paper pays attention to the generation of fine-grained small objects like books on the shelf and plates on the table, which is important for embodied AI research like object manipulation tasks.
3. The proposed method is clear and simple, which should be easy to follow. And the generated scenes seem reasonable and friendly for embodied AI tasks.
4. The limitations and future work are properly discussed. It points out that the current method retrieves furniture and large objects from datasets, limiting the diversity.

**Weaknesses:**

1. Faithfulness to image generation results? The paper seems to be using retrieval models or text-to-3D models after acquiring coarse 3D information about an object, which may be unfaithful to the appearance cue provided by the diffusion model.

2. Unfair comparison. The comparison with Text2Room does not seem very fair as this paper retrieves furniture items from carefully designed datasets while Text2Room generates furniture items from scratch. On the other hand, as the paper focuses on embodied AI tasks, PhyScene[1] can serve as a potential baseline that considers many physical plausible constraints and it would be interesting to see the comparison results.

3. Limited technical contribution. Though the paper mixes lots of off-the-shelf models, the main idea is to acquire layout from pre-trained image diffusion models, whose capability can be justified in detail.

**Questions:**

Apart from the weakness section, I still have the following questions.

1. The idea of leveraging pre-trained image diffusion models to provide a layout prior to the scene is interesting. However, how well can image diffusion models provide scene layout prior can be discussed more thoroughly. For example, the diffusion model seems to be applied from a certain perspective, lacking the awareness of the holistic scene. Will it generate semantic implausible scenes like two beds in one scene?
2. The 3D constraint seems to be a 3D bounding box generated by GPT4v, how well can it guarantee that objects will not be floating above the surface or will there be other physical constraints since a physically plausible scene is important for embodied AI tasks?
3. Is the small objects decomposable? For example, in the Living Room & Dining Room case in Fig.3, there exists a plate with multiple objects on it. Are these objects decomposable in the pipeline of GPT4v--Grounding DINO--2D to 3D? If not, will this fine-grained synthesis results hinder downstream tasks like grasping?

**Limitations:**

As discussed in the paper, they use off-the-shelf generative models or retrieved models. The diversity of retrieval results is limited and the quality and articulation structure of generative models is also limited.

---

> ### Author Rebuttal · Authors · 2024-08-06
>
> # Response to Reviewer Te9g
> *We appreciate your positive and insightful comments! Below, we address your concerns in detail.*
>
> **1. Faithfulness to image generation results.**
>
> Since our ultimate goal is to generate diverse and realistic interactive scenes, the inpainted images serve as guidance to place all the assets reasonably rather than as a strict target to align with.
>
> Furthermore, we have applied several methods to improve faithfulness, such as retrieving assets based on image similarity. The details are provided in General Response 3.
>
> We've also evaluated the image similarity between inpainted images and images of generated scenes against empty scenes, as shown in Rebuttal Table 1. The results indicate that, although not exactly the same, the generated scenes are to some degree faithful to the image generation results.
>
> **2. Comparison with baselines.**
>
> The comparison with PhyScene is shown in Rebuttal Table 2, where we present the comparison results for generated living rooms (only the weight of living room generation is released).
>
> In our experiments, we aim to provide a general comparison with three types of related works: works that generate only the static mesh [1], works that are trained on existing datasets [2] and works that generate open-vocabulary scenes using foundation models [3].
>
> It's challenging to make a completely fair comparison between our methods and the Text2Room method since they serve different purposes. While Text2Room generates the entire mesh without retrieving objects, none of its assets are interactive and it may achieve higher photorealism by directly generating meshes from 2D images.
>
> **3. Technical Contribution.**
>
> We are the first to utilize 2D diffusion models for interactive scene generation. We have also implemented a pipeline that accurately lifts 2D inpainted images to 3D space, leveraging known camera parameters and partial depth information during the rendering process.
>
> The problem of accurately acquiring the 3D information for a single generated 2D image challenging. Some previous works such as [4] try to address this problem but can't get accurate results due to the lack of information of depth scale and camera informations. We are the first to address this problem by rendering in simulation and inpainting, where we naturally get the camera informations and the scale of depth.
>
> ### Questions
>
> > **Q1: The diffusion model seems to be applied from a certain perspective, lacking the awareness of the holistic scene. Will it generate semantic implausible scenes like two beds in one scene?**
>
> An inverse problem is mentioned by reviewer YmnU.
> This is addressed in our work by:
> 1. Ensuring that later views can observe some generated assets from prior views and remain consistent during inpainting.
> 2. Using a large language model to guide the inpainting process, as mentioned in paper lines 189-193. This helps prevent inconsistencies.
>
> > **Q2: The 3D constraint seems to be a 3D bounding box generated by GPT4v, how well can it guarantee that objects will not be floating above the surface or will there be other physical constraints since a physically plausible scene is important for embodied AI tasks?**
>
> The 3D constraint is based on 3D bounding boxes from point clouds, generated through segmentation and depth estimation. We directly generate the constraints based on the distance and directional relationships between these bounding boxes; more details can be found in Appendix A of the paper.
>
> Similar to Holodeck[3], during the process of solving these constraints, we always enforce collision-free placement between furniture pieces, ensuring all furniture is either placed on the floor or hung on the wall.
>
> For small objects, we use the 3D positions of the bounding boxes. Given that the objects are all placed in reasonable locations (without severe penetration or floating), we can simply run forward physics simulations for several steps to resolve these issues.
>
> > **Q3: Is the small objects decomposable? For example, in the Living Room & Dining Room case in Fig.3, there exists a plate with multiple objects on it. Are these objects decomposable in the pipeline of GPT4v--Grounding DINO--2D to 3D? If not, will this fine-grained synthesis results hinder downstream tasks like grasping?**
>
> It's not always decomposable, given that Grounding DINO might fail to segment certain objects. For example, the fruit bowl in Figure 3 is recognized and generated as one 3D mesh. This could be mitigated by first generating an empty bowl and then performing hierarchical inpainting inside it.
>
> This limitation does not affect our downstream pipeline since we are automatically generating tasks based on the ground-truth semantics of each object in the scene. The fruit bowl will be treated as a single asset and thus we won't generate tasks like picking up the fruit from the fruit bowl.
>
> [1] Höllein, Lukas, et al. "Text2room: Extracting textured 3d meshes from 2d text-to-image models." Proceedings of the IEEE/CVF International Conference on Computer Vision. 2023.
>
> [2] Tang, Jiapeng, et al. "Diffuscene: Denoising diffusion models for generative indoor scene synthesis." Proceedings of the IEEE/CVF conference on computer vision and pattern recognition. 2024.
>
> [3] Yang, Yue, et al. "Holodeck: Language guided generation of 3d embodied ai environments." Proceedings of the IEEE/CVF Conference on Computer Vision and Pattern Recognition. 2024.
>
> [4] Chen, Boyuan, et al. "Spatialvlm: Endowing vision-language models with spatial reasoning capabilities." Proceedings of the IEEE/CVF Conference on Computer Vision and Pattern Recognition. 2024.
>
> *We sincerely appreciate your comments. Please feel free to let us know if you have further questions.*
>
> Best,
>
> Authors

---

> > ### Comment · Reviewer_Te9g · 2024-08-13
> >
> > I have read the rebuttal carefully and most of my concerns are addressed. Hence, I decide to keep my original score and am leaning to accept this paper.

---

> ### Author Response · Authors · 2024-08-13
>
> We are glad to know that you are leaning to accept this paper and that your concerns have been addressed. If there’s anything more we can do to further enhance your view of our work, we would be glad to provide additional details. Thank you again!

---

### Official Review · Reviewer_yxqm · 2024-07-12

**Soundness:** 3
**Presentation:** 3
**Contribution:** 3
**Rating:** 6
**Confidence:** 4

**Summary:**

This paper considers creating large 3d indoor scenes (e.g. an apartment or grocery store) by an hierarchical generation procedure. The key idea is to use diffusion models for inpainting to guide where to place the objects in the scene. This is done for both large objects (e.g. table or couch) and also small objects relative to the larger ones (e.g. books or a coffee cup on a coffee table). The objects are retried from both large scale object datasets (Objaverse) and image-to-3d models. The paper is compared to several existing methods for scene generation, and found to perform well, generating scenes that adhere to the text prompt. The scenes generated by the method could be useful e.g. for robotics and for training embodied agents, although such experiments are not presented.

**Strengths:**

- The paper presents a methodology to use images created by generative models (inpainting version of stable diffusion xl) to find relative object placements. This is in contrast to existing work often relying on heuristics, generative models for scene configurations, or object relations generated by text generated by LLMs.
- Qualitatively, the generated scenes are detailed, with multiple objects that look reasonably placed and with fine-grained details. Also quantitatively, experiments show e.g. higher CLIP and BLIP similarity between the generated scenes and text prompts than several works that are compared to.
- The paper presents an hierarchical approach to placing objects where it starts with an empty scene, and then large furniture is placed, and finally small objects are placed on the large objects, guided by the inpaintings.

**Weaknesses:**

- The 2d inpainting is used only partially. For the large objects, it is only used as a means of finding object placements in the form of 3d bounding boxes, and then the content within the bounding box (inpainted image, point cloud etc) is more or less discarded and an object is retrieved from Objaverse instead and placed in the 3d bounding box. This is not the case for small objects where the inpainted images are fed to an image-to-3d model, and it is unclear why this distinction between large and small objects was made.
- Several of the closest related works, with two examples being Holodeck [Yang 2024c] and procTHOR [Deitke 2022], evaluate how agents for e.g. object navigation perform after training on their proposed datasets, and evaluate how it generalizes to other datasets. Such policy learning experiments are missing from the current paper.
- Qualitatively, it clearly looks synthetic and not photo-realistic both for the scene layout with the floor and walls, and also for the large objects, which are retrieved from partially synthetic datasets. This is also the case for the closest related work (e.g. Holodeck), so it is not a major issue because of this.

**Questions:**

See weaknesses

**Limitations:**

This is adequately addressed

---

> ### Author Rebuttal · Authors · 2024-08-06
>
> # Response to Reviewer yxqm
> *We appreciate the positive and insightful comments from you! We adress your concerns in details below.*
>
> **1. Partially Usage of 2D Inpainting**
>
> Generally speaking, the occlusion when inpainting large furniture is much more severe than when inpainting small objects. Therefore, we treat them differently. But in our latest pipeline, we incorporate image similarity for object selection, as mentioned in General Response point 3A. We also want to clarify that the inpainted image is not fed into the image-to-3D pipeline, as mentioned in General Response point 3B.
>
> More details are included in General Response points 3A and 3B.
>
> **2. Policy Learning Experiments**
>
> To the best of our knowledge, only Holodeck[1] and procTHOR[2] demonstrate policy learning from their collected data, as they are based on the AI2-THOR platform, which provides comprehensive APIs and benchmarks for navigation and other semantic embodied tasks. In contrast, other related works [3, 4, 5, 6, 7, 8] do not focus on such tasks.
>
> Our work significantly advances small object placement, a task particularly relevant to robotic manipulation. Consequently, our research aligns more closely with previous works like RoboGen[9] and Gensim[10], which emphasize the acquisition of manipulation skills. Therefore, our experiments for embodied tasks are centered around manipulation skill acquisition rather than training agents on the proposed dataset.
>
> We've illustrated more details of these experiments in General Response 2A.
>
> **3. Synthetic Style**
>
> We believe that the reason some scenes still appear synthetic is mainly due to two factors: 1) the retrieved assets are not realistic enough, and 2) the rendering configuration (surface materials and lighting conditions) is not carefully adjusted.
>
> We can address these issues by retrieving assets from artist-designed datasets and improving the rendering configuration. As shown in Rebuttal Figures 1 and 3, with a better rendering configuration, our rendering results have become more photorealistic than before.
>
> [1] Yang, Yue, et al. "Holodeck: Language guided generation of 3d embodied ai environments." Proceedings of the IEEE/CVF Conference on Computer Vision and Pattern Recognition. 2024.
>
> [2] Deitke, Matt, et al. "🏘️ ProcTHOR: Large-Scale Embodied AI Using Procedural Generation." Advances in Neural Information Processing Systems 35 (2022): 5982-5994.
>
> [3] Wen, Zehao, et al. "Anyhome: Open-vocabulary generation of structured and textured 3d homes." arXiv preprint arXiv:2312.06644 (2023).
>
> [4] Yang, Yandan, et al. "Physcene: Physically interactable 3d scene synthesis for embodied ai." Proceedings of the IEEE/CVF Conference on Computer Vision and Pattern Recognition. 2024.
>
> [5] Tang, Jiapeng, et al. "Diffuscene: Denoising diffusion models for generative indoor scene synthesis." Proceedings of the IEEE/CVF conference on computer vision and pattern recognition. 2024.
>
> [6] Höllein, Lukas, et al. "Text2room: Extracting textured 3d meshes from 2d text-to-image models." Proceedings of the IEEE/CVF International Conference on Computer Vision. 2023.
>
> [7] Feng, Weixi, et al. "Layoutgpt: Compositional visual planning and generation with large language models." Advances in Neural Information Processing Systems 36 (2024).
>
> [8] Raistrick, Alexander, et al. "Infinigen Indoors: Photorealistic Indoor Scenes using Procedural Generation." Proceedings of the IEEE/CVF Conference on Computer Vision and Pattern Recognition. 2024.
>
> [9] Wang, Yufei, et al. "Robogen: Towards unleashing infinite data for automated robot learning via generative simulation." arXiv preprint arXiv:2311.01455 (2023).
>
> [10] Wang, Lirui, et al. "Gensim: Generating robotic simulation tasks via large language models." arXiv preprint arXiv:2310.01361 (2023).
>
>
> *We hope the additional explanations have convinced you of the merits of our work.*
> *We appreciate your time! Thank you so much!*
>
> Best,
>
> Authors

---

> ### Author Response · Authors · 2024-08-13
>
> Dear reviewer yxqm,
>
> We are truly grateful for your insightful comments and advice, which have played a significant role in enhancing the quality and clarity of our paper.
>
> We hope that the additional details and experimental results we provided have effectively addressed your concerns. As the rebuttal period comes to an end, we kindly request your thoughts on our rebuttal and ask that you consider raising your score accordingly. If there are any remaining concerns, please feel free to share them with us.
>
> Once again, we deeply appreciate your thoughtful review and constructive feedback.
>
> Best,
>
> Authors

---

> > ### Comment · Reviewer_yxqm · 2024-08-13
> >
> > I thank the authors for the answers which clarified my concerns, and I appreciate the experiments with embodied skill acquiring (sec. 2A in the general response post). I updated my score one step to weak accept.

---

### Official Review · Reviewer_YmnU · 2024-07-12

**Soundness:** 3
**Presentation:** 3
**Contribution:** 3
**Rating:** 7
**Confidence:** 4

**Summary:**

This paper proposes a hierarchical diffusion-based 2D inpainting method for creating interactive 3D scenes. By leveraging the generative prior of 2D diffusion models, the proposed method could generate more realistic and diverse object layouts compared with Holodeck [Yang et al., 2024c] that depends on LLMs which lack 3D spatial reasoning ability. Specifically, it renders a 3D scene (initialized with empty scene) and utilizes its GT depth and camera parameters for the back-projection from 2D inpainted components to 3D point clouds. Based on this layout, it determines how to place 3D objects into the interactive 3D scene, where large objects are retrieved from databases and small objects are generated with text-to-3D and image-to-3D generative models.

**Strengths:**

This paper has the following strengths:

(i) The proposed hierarchical diffusion-based 2D inpainting method is reasonable and seems to be effective for generating diverse and realistic object layouts. With the help of the prior encoded in 2D diffusion models, it shows better 3D spatial reasoning compared with Holodeck that heavily depends on LLMs. As shown in Figures 3 and 4, it could also synthesize and organize small objects, which is interesting.

(ii) This paper comprehensively evaluates the proposed method in terms of CLIPScore, BLIPScore, VQAScore and GPT4o Ranking, and also conducts a user study. As shown in Table 2, it achieves state-of-the-art results. The quantitative results are supported by qualitative comparisons shown in Figures 5 and 6.

(iii) This paper is well-organized and easy to understand. Most descriptions are clear.

**Weaknesses:**

This paper has the following weaknesses:

(i) The diversity of inpainting results might be affected by the bias of LLMs, sinch inpainting prompts are generated by LLMs. According to L191-193, it says “TV” will be placed into the negative prompt since a living room normally has only one TV set. Such bias might enhance the realism but limit the diversity of inpainting.

(ii) The proposed method essentially enables generating open-vocabulary scenes, but it currently retrieves large furniture pieces from databases.

(iii) Several implementation details are not fully described. For example, it does not elaborate the predefined criterion used in the filtering (L194-195) and how to finally select large furniture (L224-228).

**Questions:**

(i) The proposed method generates small objects using a text-to-3D generation method (MVdream) and an image-to-3D generation method (InstantMesh), but it retrieves large furniture pieces instead of such generation. Is this because the large furniture pieces determine how small objects are placed and thus should be neat? What is the reason for the retrieval?

(ii) According to L194-195, the proposed method filters out images if the number of recognized objects falls below a pre-defined criterion. What is the criterion? Is there also an upper bound?

(iii) According to L224-228, the proposed method finally selects large furniture pieces based on feature similarity and proportionality of their scale. How did the authors compute feature similarity and balance between the similarity and proportionality?

**Limitations:**

Yes, in page 9 and 18.

---

> ### Author Rebuttal · Authors · 2024-08-06
>
> # Response to Reviewer YmnU
> *We appreciate the positive and constructive comments from you! We have modified our paper according to your comments.*
>
> **1. Bias from LLMs**
>
> There is always a trade-off between diversity and realism. It is true that sometimes chaotic data can be valuable in enhancing an agent's capability to deal with such scenarios. We'll make this LLM bias optional in our future version. Thank you for mentioning this!
>
> **2. Retrieving from Dataset**
>
> This is a limitation of our work, as mentioned in the Limitation and Future Work section of our paper. However, we can address this limitation by augmenting our method with a 3D generation pipeline for large furniture, as shown in General Response 2C.
>
> **3. Implementation Details**
>
> We've provided the implementation details in General Response 3.
> For the predefined criterion used in the filtering, we filter out the inpainting images with fewer than three new objects. If 30 images are filtered out, we choose the one with the most objects from them.
>
> ### Questions
>
> > **Q1: The proposed method generates small objects using a text-to-3D generation method (MVdream) and an image-to-3D generation method (InstantMesh), but it retrieves large furniture pieces instead of such generation. Is this because the large furniture pieces determine how small objects are placed and thus should be neat? What is the reason for the retrieval?**
>
> We retrieve large furniture instead of generating it mainly because the quality of retrieved large assets is generally better than that of generated large assets. Conversely, for small objects, the quality of generated objects is generally higher and more diverse than retrieved ones. To address this limitation, we have developed a pipeline for generating higher quality large furniture, as shown in General Response 2C.
>
> Another reason for retrieving large furniture is our aim to make assets more interactive. We sometimes add articulated furniture to the scene, for which there is currently no suitable generation pipeline.
>
> > **Q2: According to L194-195, the proposed method filters out images if the number of recognized objects falls below a pre-defined criterion. What is the criterion? Is there also an upper bound?**
>
> The lower bound is 3 and there's no upper bound.
>
> > **Q3: According to L224-228, the proposed method finally selects large furniture pieces based on feature similarity and proportionality of their scale. How did the authors compute feature similarity and balance between the similarity and proportionality?**
>
> This is included in General Response point 3A. Essentially, we use feature similarity to retrieve multiple assets and then use proportionality to select from them.
>
> *We hope the additional explanations have convinced you of the merits of our work. Please do not hesitate to contact us if you have other concerns.*
>
> *We really appreciate your time! Thank you!*
>
> Best,
>
> Authors

---

> ### Comment · Reviewer_YmnU · 2024-08-10
>
> Dear Authors,
>
> Thanks for the response! My concerns were addressed in a proper way.
>
> I think this paper is worthwhile to be accepted.
>
> Best regards,
>
> Reviewer YmnU

---

### Official Review · Reviewer_YpUS · 2024-07-13

**Soundness:** 2
**Presentation:** 3
**Contribution:** 2
**Rating:** 5
**Confidence:** 4

**Summary:**

The paper introduces ARCHITECT, a generative framework designed to create complex and realistic 3D environments for Robotics and Embodied AI research. Unlike traditional methods that rely on manual design, predefined rules, or large language models (LLMs), ARCHITECT utilizes pre-trained 2D image generative models to capture scene and object configurations more effectively. To address the problem of camera parameters and depth estimation, ARCHITECT utilizes a hierarchical inpainting pipeline.
Specifical contributions are: 1. The paper proposes to leverage 2D prior from vision generative models for the 3D interactive scene generation process to generate complex and cluttered object arrangements; 2. The paper introduces a controlled inpainting process that utilizes a controllable rendering process and integrates geometric cues from the background to guide the generation of the foreground, allowing for better control over camera parameters and depth perception.

**Strengths:**

1. Compared to previous work that overly relies on the ability of LLMs, this work proposes the use of inpainting models to draw room layouts, making the generated furniture layouts more diverse. This can inspire future work to better utilize different pre-trained multimodal models to assist in generation, control, and other task.
2. This work proposes a solution for the arrangement of small objects, which helps generate more realistic scenes and is more conducive to subsequent interaction planning.

**Weaknesses:**

1. For methods based on rules and LLMs, we can directly edit the positions, types, and other conditions of objects to control the output. However, when using a pre-trained inpainting model to plan object layouts, it becomes difficult to make such detailed edits, which seems to result in a loss of some controllability.
2. Although the goal of the paper is to generate scenes using multiple viewpoints, the paper only provides the method of selecting the first camera viewpoints. I hope the authors can further clarify the specific rules for selecting multiple viewpoints.
3. Heuristically choosing the view that spans from one corner to the opposite corner results in only half of the room being occupied by objects, which would make the room appear unrealistic.

**Questions:**

1. Can the inpainting model ensure that the appearance of the masked area is consistent with other areas?
2. Could you provide some results about generating using image-to-3D generative models? I found that the paper mentioned this approach in the introduction section but did not provide subsequent experiments.
3. How do you perform object retrieval? Is it like the HOLODECK method?
4. Can the algorithm based on LLMs solve the problem of small object arrangement by modifying the example in the prompt?

**Limitations:**

The authors have addressed the limitations. However, the limitations of inpainting models still need to be further elucidated.

---

> ### Author Rebuttal · Authors · 2024-08-06
>
> # Response to Reviewer YpUS
>
> *Thank you for your insightful and constructive comments! We have added additional experiments and modified our paper according to your comments.*
>
> **1. Controllability and Editing**
>
> In short, our method combines a diffusion-based pipeline with an LLM-based method, which still possess the ability of controlling and editing. The inpainting-to-layout pipeline functions can be considered as an API function callable by the LLM. Our approach aims to generate scene configurations seeded from diffusion models, with scene editing as an othogonal feature enabled by LLMs.
>
> Specifically, the scene configuration generated by our pipeline can be represented by each object's name, position, scale, bounding box, orientation, and asset uid, which can be easily converted to text representations. This allows us to feed this information directly into LLMs to further control or edit the scene layout.
> The corresponding experiments could be found at General Response 2B.
>
> **2. View Selection**
>
> As illustrated in General Response section 3C, we heuristically select up to three views to ensure that most areas of the room are inpainted and appropriately placed.
>
> **3. Only Half Room Inpainted With a Single View**
>
> Addressing this concern, we select up to three different views to cover the entire room area for inpainting. Additionally, we use an 84-degree FOV for our camera during rendering, a standard parameter for real-world cameras. Consequently, for a square room, this setup results in approximately 95 percent of the room being visible from a single corner-to-corner view.
>
> ### Questions
>
> > **Q1: Can the inpainting model ensure that the appearance of the masked area is consistent with other areas?**
>
> Yes, the appearance of the masked area is consistent with other areas both stylistically and geometrically. We also apply a commonly used technique, softening the boundary of inpainting masks, to improve consistency. A comparison of the results before and after using softened inpainting masks is shown in the left part of Rebuttal Figure 3.
>
> > **Q2: Could you provide some results about generating using image-to-3D generative models? I found that the paper mentioned this approach in the introduction section but did not provide subsequent experiments.**
>
> We are using the previous work InstantMesh[1] to generate 3D assets. This is not our main contribution and is not specifically highlighted in our paper. Some of the results can be seen in the images within our paper, where all the small objects are generated using InstantMesh. Additionally, we present example generated 3D assets in Rebuttal Figure 2, which also includes new results from our large object generation pipeline.
>
> > **Q3: How do you perform object retrieval? Is it like the HOLODECK method?**
>
> As illustrated in General Response 3A and 3B, our method is basically the same as the Holodeck[2] retrieving method.
>
> > **Q4: Can the algorithm based on LLMs solve the problem of small object arrangement by modifying the example in the prompt?**
>
> It's challenging for LLMs to directly solve arrangement problems. First, for small object placement on shelves, LLMs lack information about supporting surfaces, making it impossible for them to solve this issue. Second, for placement on tables, while we might know the supporting surfaces given the bounding boxes, LLMs struggle with object orientations, often resulting in less complex scenes or scenes with severe collisions.
> We show a comparison of small objects generated by our methods and LLMs in the middle part of Rebuttal Figure 3 and in Table 1.
>
> [1] Xu, Jiale, et al. "Instantmesh: Efficient 3d mesh generation from a single image with sparse-view large reconstruction models." arXiv preprint arXiv:2404.07191 (2024).
>
> [2] Yang, Yue, et al. "Holodeck: Language guided generation of 3d embodied ai environments." Proceedings of the IEEE/CVF Conference on Computer Vision and Pattern Recognition. 2024.
>
> *We wish that our response has addressed your concerns. If you have any more questions, please feel free to let us know during the rebuttal window.*
>
> Best,
>
> Authors

---

> > ### Comment · Reviewer_YpUS · 2024-08-13
> >
> > Thanks for the effort in the rebuttal. My concerns are resolved to a large extent so my score will remain unchanged.

---

> ### Author Response · Authors · 2024-08-13
>
> Dear reviewer YpUs,
>
> Thank you once again for your insightful comments and advice, which have been instrumental in improving the quality and clarity of our paper.
>
> As the rebuttal phase is drawing to a close, we are eager to hear your thoughts on our response. We hope our additional details and experimental results have addressed your concerns. We kindly ask if you could provide your opinion on our rebuttal and adjust your score accordingly. If you have any additional concerns, please do not hesitate to share them with us.
>
> Once again, we sincerely appreciate your thoughtful review and constructive feedback.
>
> Best,
>
> Authors

---

### Author Rebuttal · Authors · 2024-08-06

# General Response to All Reviewers
*We express our gratitude to all the reviewers for their perceptive comments and helpful suggestions aimed at enhancing the quality of our work.*

**1. Our Contributions**

We are pleased that the reviewers have generally acknowledged our contributions:

* We properly leverage 2D priors from vision generative models to generate 3D interactive scenes.
* We implement a hierarchical inpainting pipeline that can 'control' the inpainting process, allowing for accurate 2D to 3D lifting.
* Our method demonstrates a strong ability to generate reasonable 3D interactive scenes compared to previous works, especially in the detailed placement of small objects.

**2. New Experiments**

In this rebuttal, we add several more supporting experiments to address reviewers’ concerns, and we select three of them to list in the General Response.

* **[A] Embodied Skill Acquiring.** We further tested our methods based on previous work, RoboGen. Specifically, we aimed to automatically generate and solve manipulation tasks using scenes generated by our pipeline. Taking the scene configuration as input, we used GPT-4 to generate tasks, decompose them, and solve them by calling primitive functions we provided. We can now generate more diverse and longer-term tasks, such as *put the mango in the fridge and move the soda from the fridge to the dining table*, which involves multiple rooms. The comparison of diversity is shown in Rebuttal Table 1 by the self-BLEU score of task discription, and some example tasks are shown in Rebuttal Figure 1.

* **[B] Scene Editing.** To demonstrate that our pipeline is compatible with scene editing and complex text control, we implemented additional APIs to add, remove, and rescale objects, enabling LLMs to edit the scene. Initial results for scene editing are shown in the right part of Rebuttal Figure 3. We issued commands to LLMs such as *replace the books on the shelf with vases*, *replace the bookshelf with a cabinet*, and *make the bookshelf smaller*. The LLMs achieved the correct results by calling the provided APIs.

* **[C] Large Furniture Generation.** To address one of our limitations, the dependence on a large furniture database, we apply a pipeline to generate high-quality large furniture. It optimizes a differentiable tetrahedron mesh [1] with SDS loss, using the normal-depth diffusion model and albedo diffusion model provided by RichDreamer[2] as the main supervision signal. This pipeline is capable of generating high-quality object meshes from text guidance, specifically large furniture in our case. Some results are shown in the right part of Rebuttal Figure 2.


**3. Implementation Details**

We observe that most of questions arise from the implementation details [YpUS, YmnU, yxqm, 4D8M]. Here, we give some detailed description of how the pipeline works.

* **[A] Large Furniture Retrieving.** Following Holodeck, for each piece of large furniture, we first retrieve multiple candidates from the dataset using text descriptions of the assets. Then, we select one asset from the retrieved candidates based on scale similarity, which is calculated as the L1 difference between the scale of 3D bounding box of object point cloud and the 3D bounding box of object mesh. Additionally, we integrated image similarity using the cosine similarity of CLIP features in the selection process in our latest pipeline. Here, scale similarity and image similarity are used only in the candidate selection process rather than the retrieval process, since there could be significant occlusions in the image (e.g., a chair behind a table) that could greatly influence the accuracy of retrieving.

* **[B] Small Objects Generation and Selection.** As also mentioned in our paper (line 255), we use a text-to-3D pipeline (text-to-image and image-to-3D) to generate 3D assets for small objects. To make the scene more reasonable and resemble the inpainted image, we generate multiple candidates for each type of object and use the cosine similarity of DINO features to select from the candidates. We also experimented with an image-to-3D pipeline, starting from the object image segmented from the inpainted image. However, the resolution of the segmented image is low, resulting in sub-optimal 3D shapes and textures.

* **[C] View Selection.** For large furniture placement, we heuristically select up to three views (right-back corner to left-front corner, front middle to back middle, and left-back corner to right-front corner) that can cover the whole room area for inpainting. Assuming the room ranges from $(0, 0)$ to $(x, y)$, the three views would be looking from $(x, y, 1.8)$ to $(0, 0, 0.5)$, from $(\frac{x}{2}, 0, 1.8)$ to $(\frac{x}{2}, y, 0.5)$, and from $(0, y, 1.8)$ to $(x, 0, 0.5)$. We stop inpainting from new views when the occupancy of the room is larger than 0.7 or it has been inpainted from all three views.
For small object placement, we first ask LLMs to determine which objects can accommodate small objects on or in them, and then inpaint each of them with heuristic relative views. For objects like tables or desks on which we are placing items, we use a top-down view. For shelves or cabinets in which we are placing objects, we use a front view. The distance of the camera from the object is adjusted according to the scale of the object and the camera's FOV, ensuring the full object is visible during inpainting.

We hope our responses below convincingly address all reviewers’ concerns. We thank all reviewers’ time and efforts again!

[1] Guo, Minghao, et al. "TetSphere Splatting: Representing High-Quality Geometry with Lagrangian Volumetric Meshes." arXiv preprint arXiv:2405.20283 (2024).

[2] Qiu, Lingteng, et al. "Richdreamer: A generalizable normal-depth diffusion model for detail richness in text-to-3d." Proceedings of the IEEE/CVF Conference on Computer Vision and Pattern Recognition. 2024.

---

### Author Response · Authors · 2024-08-11
**Thank you and we are looking forward to your post-rebuttal feedback!**

Dear AC and all reviewers:

Thanks again for all the insightful comments and advice, which helped us improve the paper's quality and clarity.

The discussion phase has been on for several days and we are still waiting for the post-rebuttal responses.

We would love to convince you of the merits of the paper. Please do not hesitate to let us know if there are any additional experiments or clarification that we can offer to make the paper better. We appreciate your comments and advice.

Best,

Authors

---

### Decision · Program_Chairs · 2024-09-25

**Decision:**

Accept (poster)

**Comment:**

The submission initially received mixed reviews; the authors did a great job during the rebuttal, after which all reviewers became positive about the submission.  The AC agrees with the recommendations.  The authors should incorporate the rebuttal into the camera ready.